# Wear and Corrosion Resistance of Plasma Electrolytic Oxidation Coatings on 6061 Al Alloy in Electrolytes with Aluminate and Phosphate

**DOI:** 10.3390/ma14144037

**Published:** 2021-07-19

**Authors:** Zhenjun Peng, Hui Xu, Siqin Liu, Yuming Qi, Jun Liang

**Affiliations:** 1College of Petrochemical Engineering, Lanzhou University of Technology, Lanzhou 730050, China; zhenjunpeng2010@licp.cas.cn; 2State Key Laboratory of Solid Lubrication, Lanzhou Institute of Chemical Physics, Chinese Academy of Sciences, Lanzhou 730000, China; liusiqin18@mails.ucas.edu.cn (S.L.); qym87@sina.com (Y.Q.)

**Keywords:** aluminum alloy, plasma electrolytic oxidation, phosphate, aluminate, wear, corrosion

## Abstract

Phosphate and aluminate electrolytes were used to prepare plasma electrolytic oxidation (PEO) coatings on 6061 aluminum alloy. The surface and cross-section microstructure, element distribution, and phase composition of the PEO coatings were characterized by SEM, EDS, XPS, and XRD. The friction and wear properties were evaluated by pin-on-disk sliding tests under dry conditions. The corrosion resistance of PEO coatings was investigated by electrochemical corrosion and salt spray tests in acidic environments. It was found that the PEO coatings prepared from both phosphate and aluminate electrolytes were mainly composed of α-Al_2_O_3_ and γ-Al_2_O_3_. The results demonstrate that a bi-layer coating is formed in the phosphate electrolyte, and a single-layered dense alumina coating with a hardness of 1300 HV is realizable in the aluminate electrolyte. The aluminate PEO coating had a lower wear rate than the phosphate PEO coating. However, the phosphate PEO coating showed a better corrosion resistance in acidic environment, which is mainly attributed to the presence of an amorphous P element at the substrate/coating interface.

## 1. Introduction

The awareness of sustainable development and environmental stability has been deeply rooted in various fields in recent decades. Accordingly, the target of reducing the consumption of natural resources and decreasing pollution has been established. As a consequence, aluminum (Al) alloys have gained more and more attentions due to its characteristic of light weight. They are considered as desirable materials to replace the traditional ferrous materials to some extent. In terms of the properties, Al alloys are characterized by high specific strength and low specific weight compared with conventional ferrous material, wide adaptability of heat treatment, and superior processability for casting, forging, welding, and machining [1,2,3]. Up to now, they have been widely employed as structural components in the fields of spacecraft, aircraft, automotive, and 3C products [4,5,6]. However, Al and its alloys are susceptible to be corroded both in the acidic and the alkaline environments, which is also a fatal shortcoming for its natural oxide film. Moreover, the poor wear resistance limits their large-scale applications [7,8].

For enhancing the combination properties of Al alloys, a large number of surface treatments have been developed, such as anodizing [9], thermal oxidation [10], vapor deposition [11], and plasma electrolysis [12]. Among these surface treatments, plasma electrolytic oxidation (PEO) is considered as an effective process to fabricate ceramic oxide coatings on Al alloys. The PEO coatings have superior mechanical properties and chemical stability that can protect the Al alloy substrate from corrosion and wear failures [13,14]. It is known to all that PEO is a multifactorial process. The electrolyte component plays a critical role, affecting the surface composition, morphology, structure, and properties of PEO coatings. Generally, the PEO electrolytes are alkaline silicate [15], phosphate [16], and aluminate solutions [17], according to their main components.

Many works reported that the PEO coatings prepared from silicate electrolyte were composed of silica, alumina, and mullite [18,19,20,21,22]. In contrast, the PEO coatings prepared from phosphate and aluminate were mainly composed of alumina compounds [23,24,25,26], although a trace of phosphorus was usually detected in the PEO coatings prepared in phosphate-based electrolyte. Additionally, the alumina also endowed the relevant PEO coatings with a better anti-corrosion and anti-wear performance. However, few works have been made on a comparison between the alumina-dominant coatings fabricated in the phosphate-based and the aluminate-based electrolytes. In this work, the effects of electrolytic components on the features of the PEO coatings were researched, and the tribological properties and acidic corrosion resistance was measured.

## 2. Experimental Details

### 2.1. PEO Process

Rectangular samples of 6061 Al alloy (weight fraction: 0.88% Mg, 0.37% Fe, 0.17% Mn, 0.5% Si, 0.2% Cu, 0.25% Zn, and balance Al) with a dimension of 20 × 20 × 4 mm^3^ were used as substrates. The aluminum plates were ground and polished using waterproof abrasive papers, ultrasonically degreased in acetone, and cleaned with distilled water. The PEO process was carried out under 200 KW bi-polar pulsed power supply (JHMAO200H, Golden Arc Green Protection Technology Development Co. Ltd., Beijing, China) with the frequency of 175 Hz and duty cycle of 30%. The schematic diagram of the current-voltage waveform is shown in Figure 1. The ancillary devices included a stainless cooling pipe, an electromagnetic stirrer, and electrolytic cell. The two different solutions were used in the PEO process: (1) 20 g/L Na_3_PO_4_·12H_2_O, 1 g/L KOH with pH about 12.44; (2) 15 g/L Na_2_AlO_2_, 1 g/L KOH with pH about 12.51. In the PEO process, Al alloy plates were treated to different time lengths (30 min and 60 min) under the constant anode voltage (500 V) and cathode voltage (80 V). During the PEO process, the electrolyte temperature was kept below 25 °C by the water-cooling system and mechanical stirring. The coated samples were washed by distilled water and air dried. The PEO coatings prepared at different treating times of 30 min and 60 min in phosphate and aluminate electrolyte were denoted as P30, P60, Al30, and Al60, respectively, in the following text.

### 2.2. Surface Characterization of the Coatings

A scanning electron microscope (SEM, JEOL, Tokyo, Japan, JSM-5601LV) was employed to observe the morphologies of the surface and cross-section of the PEO coatings. The element distribution was studied using energy-dispersive X-ray spectrometer (EDS, Oxford Instruments, Oxon, England, X-Max^N^ 80). X-ray photoelectron spectroscopy (XPS, ThermoFisher Scientific, Waltham, MA, USA, ESCALAB 250Xi) as a surface analysis technique, equipped with a standard Al Ka radiation (1486.6 eV), was used to mensurate the surface composition. The phase composition was analyzed by X-ray diffractometer (XRD, Rigaku, Tokyo, Japan, D/Max-2400) using Cu Kα radiation from 20° to 80°. The thickness of the PEO coatings was measured by a digital coating thickness gauge (Elektro Physik, Köln, Germany, Mini-test 1100).

The hardness was measured on a micro-hardness tester (Hengyi Science and Technology Corporation, Shanghai, China, MH-5-VM) with a load of 500 g, pressing 10 s and holding 5 s. The friction coefficient was performed using a UMT-TriboLab (Bruker, Billerica, MA, USA) reciprocating friction and wear tester in a ball-on-disk contact configuration with the Si_3_N_4_ ball as the grinding material. The applied loads ranged from 5 N, 7 N, to 10 N at a frequency of 5 Hz with a reciprocating stroke of 10 mm. The testing time was set to 1800 s with room temperature at 25 °C and relative humidity at 30%.

The potentiodynamic polarization of the samples acidic 5 wt % NaCl solution (pH = 3.1–3.3, we added an amount of glacial acetic acid to the salt solution to ensure that the pH is between 3.1 and 3.3; see the following text.) were measured by an Autolab PGSTAT302N electrochemical workstation, employing a three-electrode configuration, with the samples as the work electrode, an Ag/AgCl electrode as the reference electrode, and a platinum foil as the counter electrode, respectively. The potentiodynamic polarization tests were scanned from −0.85 to 0.85 V at a scanning rate of 0.01 mV/s with the reference to the open circuit potential (OCP).

The anti-corrosion property tests were carried out on the salt spray tester in accordance with the standard ISO 9227–2006 with acidic 5 wt % NaCl solution (pH = 3.1–3.3). The samples were placed 45°, their surface states were recorded every 24 h, and the surface changes were recorded by a digital camera. In order to ensure the reproducibility, three specimens for each coating were used in the acid salt spray tests.

## 3. Results and Discussions

### 3.1. Morphologies of the Coatings

Figure 2 displays the surface morphologies of the PEO coatings prepared from phosphate and aluminate electrolytes. It can be seen from the figure that there are micro-pores, pancake-like structures, and micro-cracks on the surface of the coatings. However, the electrolyte composition has a great effect on the surface morphology. For the coating formed in the phosphate electrolyte, as can be seen from the Figure 2a, many irregular pores and small cracks uniformly distribute on the P30 coating surface, and most of the pancakes show an overlapping state on the P60 coating (Figure 2b). The formation of micro-pores can be attributed to the residual plasma discharge channels, and the pancake-like structures are caused by molten oxide ejected from the discharge channels owing to the rapid solidification under the cooling electrolyte. In the aluminate electrolyte, a dense surface with clustered particles and micro-cracks are obtained, yet some micro-pores also distribute around the pancakes, as shown in Figure 2c,d.

In addition, the surface characteristics of PEO coatings also change observably with treating time. For 30 min treatment, the PEO coatings exhibit porous and flat structures, as shown in Figure 2a,c. When the treatment time increases to 60 min, the pancake structures with diameters more than 15 μm and micro-cracks appear on the surface. The surface of the P60 coating is compact and uniform (Figure 2b), while there are a few more clustered particles around the pancake on the surface of the Al60 coating (Figure 2d). It should be noted that each pancake has a closed pore in the center, which is closely relevant to the gas bubbles and the plasma discharge during the PEO process. Obviously, the surface morphology of the P60 and Al60 coatings gradually becomes rough and compact, and the number of micro-cracks also increases accordingly. The formation of micro-cracks could be caused by thermal stress on the solidification process of the molten oxide and the different phase expansion coefficients of the coating [27,28].

### 3.2. Cross-Sectional Morphologies and Elemental Distributions of the Coatings

Figure 3 shows the cross-sectional morphologies and elemental distributions of the PEO coating prepared in phosphate and aluminate electrolytes for different time points. The average thicknesses of the P30, Al30, P60 and Al60 coatings are ≈21 μm, ≈22 μm, ≈31 μm, and ≈32 μm, respectively. As shown in Figure 3a, the P30 coating shows a bi-layered structure. The outer layer of the PEO coatings has a number of pores, while the inner layer is relative dense with a thickness of ≈3 μm at the Al alloy substrate and PEO coating interface. For the Al30 coating, it is relatively dense, and there are no pores in the coating, as shown in Figure 3c. With the treating time prolonged, there is a little change in the structure of the coating. For the P60 coating, the size of pores is bigger and the thickness of the inner layer increases to 5–10 μm (Figure 3b). However, the Al60 coating shows a compact single-layered coating structure. The corresponding EDS elemental mappings shown in Figure 3e–g indicate that there is obvious enrichment of P element in the denser inner layer of the P60 coating, while the porous outer layer is almost completely composed of Al and O elements. Interestingly, the P element mainly distributes at the substrate/coating interface, which forms a phosphorus-rich dense barrier layer. The Al60 coating exhibits a uniform distribution of Al and O elements along the entire coating, as show in Figure 3h,i.

### 3.3. Phase and Chemical Composition of the Coatings

Figure 4 presents the XRD patterns of PEO coatings prepared from phosphate and aluminate electrolytes at different time points. It can be seen that the characteristic diffraction peaks corresponding to α-Al_2_O_3_ (JCPDS NO. 01-1243), γ-Al_2_O_3_ (JCPDS NO. 10-0425), and Al are clearly identified for the P30, Al30, P60, and Al60 coatings. In addition, there is a hump between 24° and 42° in 2θ for the P30 and P60 coatings, indicating the presence of amorphous compounds. It is well known that alumina phases are different not only in their structure but also in the properties. Usually, the α-Al_2_O_3_ phase exhibits higher hardness compared with the γ-Al_2_O_3_ phase. The γ-Al_2_O_3_ to α-Al_2_O_3_ transition occurs at temperatures above 950 °C. During the PEO process, the temperature of the spark discharge can reach 3000 °C or higher, which promotes the transformation of the γ-Al_2_O_3_ phase to the α-Al_2_O_3_ phase. Furthermore, the low thermal conductivity of aluminium oxide results in the underlayer of the coatings to become heated and also facilitates the further transformation of the initially formed metastable γ-Al_2_O_3_ phase into the stable α-Al_2_O_3_ phase. The relative content of the α-Al_2_O_3_ and γ-Al_2_O_3_ phases in PEO coatings can be calculated from the integrated intensities of the α-Al_2_O_3_ (113) and γ-Al_2_O_3_ (400) peaks, Iα and Iγ, respectively [17,29,30,31]. The ratios of Iα/Iγ for the P30, Al30, P60, and Al60 coating are 0.01, 0.47, 0.12, and 1.64, respectively. Therefore, the relative content of α-Al_2_O_3_ is significantly increased for the PEO coatings with increasing treatment time, and the ratio of Iα/Iγ of aluminate PEO coatings is higher than that of phosphate PEO coatings at the same treatment time, indicating that the aluminate electrolyte more easily forms the α-Al_2_O_3_ phase in the PEO coatings than the phosphate electrolyte.

Figure 5a shows the XPS full-survey spectra of the P60 and Al60 coatings. The P60 coating contains Al, O, and P elements, and the Al60 coating consists of Al and O elements, which are identical with the results of EDS analyses. The C element is the impurities on surface of the coating. The specific Al 2p spectra show peaks at binding energies of 74.8 eV and 74.3 eV that correspond to Al_2_O_3_, as shown in Figure 5b [12,32]. Furthermore, the peaks of PO_3_^−^ with binding energy at 134.5 eV for P30 and P60 coatings illustrate that metaphosphate presents on the P60 coating, as shown in Figure 5c [33], which indicates that the hump in the XRD patterns of the P30 and P60 coatings are attributed to the presence of amorphous metaphosphate.

### 3.4. Micro-Hardness of the Coatings

Figure 6 shows the micro-hardness of 6061 Al alloy substrate and PEO coatings obtained from different electrolytes with different treating time. As we can see from the graph, the micro-hardness of 6061 Al alloy substrate is about 155 HV. For the PEO treatment, the micro-hardness is significantly enhanced. The Al60 coating has the largest average hardness of 1300 HV, which is 8 times that of the 6061 Al alloy substrate. Nevertheless, the micro-hardness of the P30 coating is relatively low and about 700 HV, which is 4 times that of the substrate. In addition, it can be deduced from Figure 5 that the micro-hardness of PEO coatings grown in the same electrolyte increases with the increase of the treatment time; that is, the greater the thickness of the PEO coating, the higher the hardness.

It is known that the micro-hardness of the PEO coatings is related to the surface structure and phase composition [28]. On the basis of the SEM results, the P60 and Al60 coatings are thicker and denser than the P30 and Al30 coatings. On the other hand, the XRD results shows that the PEO coatings prepared from aluminate electrolytes with longer treating times have a higher content of α-Al_2_O_3_ phase, which is helpful for improving the hardness.

### 3.5. Friction and Wear Properties of the Coatings

Figure 7 presents the friction curves of P30, Al30, P60, and Al60 coatings at 5 N loads under dry friction conditions. As we can see from the curves, the friction coefficient of the PEO coatings fluctuates with the sliding time. There is a run-in stage in the friction process, which may be concerned with the irregular surface morphology of the coatings [28]. The run-in stages of all the samples are about 200 s, and the friction coefficient became stable within the range from 0.75 to 0.85, as observed in Figure 7. For the P30 coating samples, the friction coefficient increases gradually to 0.78 within 960 s and then drops suddenly, indicating that the coating has been worn through. For the Al30, P60, and Al60 coatings, they have a relatively high friction coefficient, which may be attributed to the larger shear stress at dry friction conditions and uneven distribution of surface microstructures, as shown in Figure 2. In addition, these PEO coatings still exist after a wear test for 1800 s, indicating that those coatings have excellent wear resistance.

Figure 8 shows the morphologies of a wear track after dry sliding tests under 5 N load. As we can see from Figure 8a, the P30 coating shows obvious tearing, plastic flow, adhesion, and furrow characteristics, indicating that the P30 coating have been thoroughly worn through. As shown in Figure 8b–d, most of the coating’s areas did not appear to have much damage, which indicated the good wear resistance of these coatings (just as the coatings test with loads of 7 N, 8 N, 9 N, and 10 N). The magnified image of morphology is presented on the upper right of Figure 8b–d. As we can see from that, the micro-morphologies of the wear track for Al30 and P60 coatings exhibit furrows and a small range of exfoliation. This illustrates that the main wear mechanisms of the PEO coatings are abrasive wear and fatigue wear. Nevertheless, the furrows and the area of exfoliation for the Al60 coating reduced in size, which is attributed to the lower porosity, higher compactness, and lack of defects in the PEO coatings.

The cross-sectional profiles of the PEO coatings are shown in Figure 9. These profiles are carried out to estimate the wear rate. The summary of the friction and wear test results of the PEO coatings is listed in Table 1. The wear tracks of the P30 coating are around 29 μm in depth and 400 μm in width. The wear track depth of the Al30 and P60 coatings are substantially reduced, which register 20 μm and 18 μm. For the Al60 coating, the values of the depth are further reduced than those of the other coatings. Benefitting from the larger thickness and higher ratio of harder compounds, there is an evident improvement of the wear resistance. Based on the wear profiles, the specific wear rates of the P30, Al30, P60, and Al60 at 5 N loads are 6.46 × 10^−4^ mm^3^/N·m, 1.37 × 10^−4^ mm^3^/N·m, 1.66 ×10^−4^ mm^3^/N·m, and 8.24 × 10^−5^ mm^3^/N·m, respectively. It is deduced that the wear rate of the Al60 coating is nearly an order of magnitude lower than that of the P30 coating.

The sliding tests of the PEO coatings at higher load (7N) were used to further evaluate the difference of their wear resistance. It is found from Figure 10 that the P30, Al30, and P60 coatings are worn through after ≈100 s, 340 s, and 440 s with loads of 7 N, respectively. However, the friction curve of the Al60 coating remains stable during the sliding time of 1800 s, which shows much better wear resistance than other coatings under the load of 7 N. To further evaluate the friction and wear property of the Al60 coating, the loads of 8 N, 9 N, and 10 N are used for the slide test, and the friction coefficient curves are shown in Figure 11. As we can see from that, the Al60 coating shows a stable friction coefficient within the sliding time of 600 s under 8 N loads. However, the friction coefficient drops at the sliding time of about 200 s under 9 N loads, while the friction curve dramatically decreased on applying a load of 10 N.

### 3.6. Corrosion Behavior of the PEO Coatings

Figure 12 displays the potentiodynamic polarization curves that were carried out in the 5 wt % NaCl (pH = 3.1–3.3) corrosive media after OCP stabilization. The corrosion potential (E_corr_) and current density (i_corr_) of the 6061 Al alloy and PEO coatings obtained from Figure 12 are listed in Table 2. As we can see from Figure 12, the PEO coatings have much lower i_corr_ and more positive E_corr_ than the Al alloy substrate, indicating that the PEO coatings improve the corrosion resistance of Al alloy. For the PEO coatings, the E_corr_ has a positive shift, and the i_corr_ decreases with the treatment time increasing. According to Table 2, the P60 coating shows the lowest corrosion current density (1.09 × 10^−7^ A·cm^−2^). Therefore, the P60 coating exhibits the best corrosion resistance compared to those other coatings in acidic solution according to the potentiodynamic polarization results.

Figure 13 presents the optical photographs of 6061 Al alloy, P60 coating, and Al60 coating after the acid salt spray test for 696 h. As we can see from Figure 13a–c, the Al alloy is severely corroded and completely covered by corrosion products after 72 h of testing, indicating that the Al alloy has low corrosion resistance in an acidic corrosion environment. The P60 coating and Al60 coating show excellent corrosion resistance than the 6061 Al alloy. For the P60 coating, no corrosion can be observed from Figure 13d–f after a 696 h acid-salt-spray test. In the case of the Al60 coating, a few corrosion pits appear on the surface after 216 h tests. Severe corrosion occurred after 696 h, suggesting that the Al60 coating lost the protection to Al alloy from corrosion to some extent, as shown in Figure 13g–i. The results indicate that the phosphate PEO coating shows much better corrosion protection performance in an acidic corrosion environment than the aluminate PEO coating.

These performance characteristics may be connected with the microstructure and composition of the PEO coatings. The P60 coating has a bi-layered structure. Although there are pores in the outer layer, the perforated pores have been blocked by that which solidified under the cooling electrolyte effection, resulting in the pores not coming through the coating [34]. Furthermore, the inner layer with the thickness of ≈10 µm of the substrate/coating interface is relatively dense, and a phosphorus-rich oxide was characterized, as shown in Figure 6. This dense layer played a vital role in the inhibition of the penetration of the acid corrosive medium. When the acid corrosive medium penetrated into the coating, the loose layer can act as a barrier to prevent the entry and decrease the rate of diffusion of acid corrosive medium [35]. Moreover, the amorphous phosphorus-rich oxide will improve the acid anti-corrosion properties of the PEO coatings, while the inner layer of the Al60 coating was mainly composed of aluminum oxide. The crystal boundaries or microcracks of the aluminum oxide are easily eroded by acid corrosive medium, which create the path of access by acid corrosive medium, resulting in the deterioration of the corrosion resistance.

## 4. Conclusions

PEO coatings were fabricated on 6061 aluminum alloy in phosphate and aluminate electrolytes, respectively, and the wear and acid corrosion resistance were compared. The conclusions are as follows: γ-Al_2_O_3_ and α-Al_2_O_3_ are the main crystal phases of the PEO coatings, accompanied by a small amount amorphous of phosphorus-rich oxide formed in phosphate electrolyte. With the increase of PEO treating time, the ratios of Iα/Iγ increased and the surface hardness increased.The single-layered coatings formed in the aluminate electrolyte exhibit more desirable wear resistance than that in the phosphate electrolyte, which is relatively high α-Al_2_O_3_ content.The bi-layered coatings formed in the phosphate electrolyte have better excellent acid corrosion resistance, which is attributed to the dense phosphorus-rich oxide of the inner layer.

## Figures and Tables

**Figure 1 materials-14-04037-f001:**
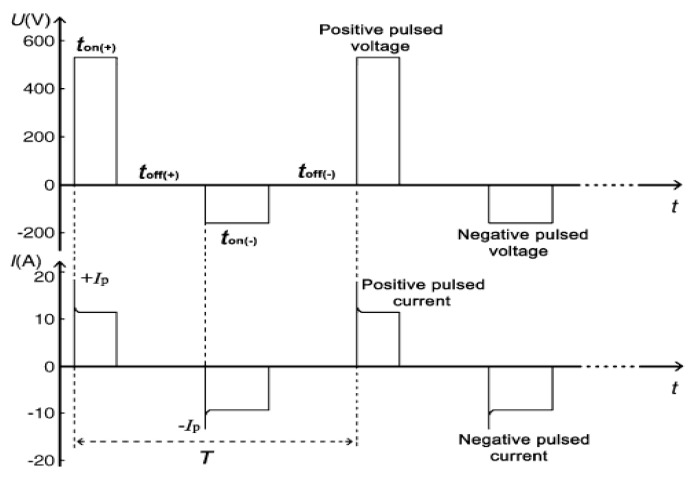
The schematic drawing of the current-voltage waveform in the process of power supply operation.

**Figure 2 materials-14-04037-f002:**
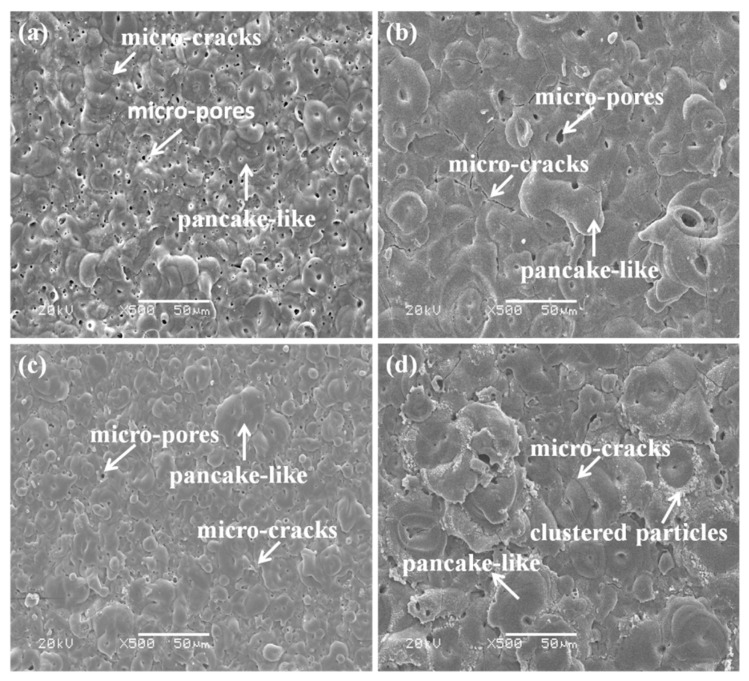
SEM morphologies of the surface coatings prepared from different electrolytes with different treating time: (**a**,**b**) P30 and P60 coatings; (**c**,**d**) Al30 and Al60 coatings.

**Figure 3 materials-14-04037-f003:**
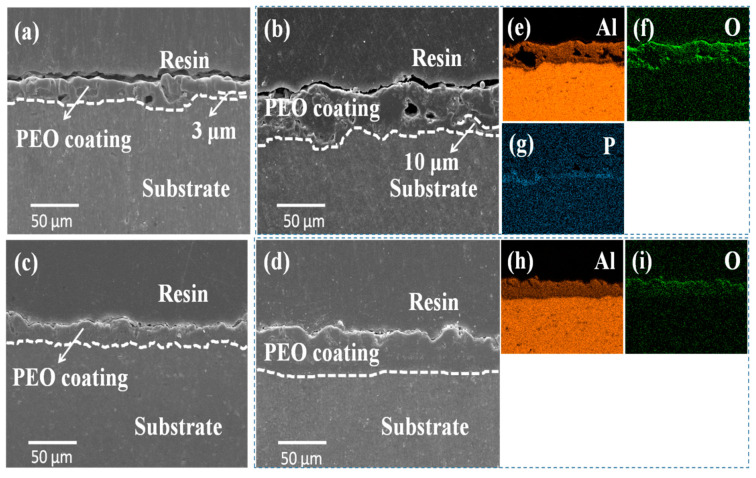
Cross-sectional SEM morphologies of the surface coatings prepared from different electrolytes with different treating time: (**a**,**b**) P30 and P60 coatings; (**c**,**d**) Al30 and Al60 coatings; (**e**–**g**) the EDS elemental mappings of P60 coating; (**h**,**i**) the EDS elemental mappings of Al60 coating.

**Figure 4 materials-14-04037-f004:**
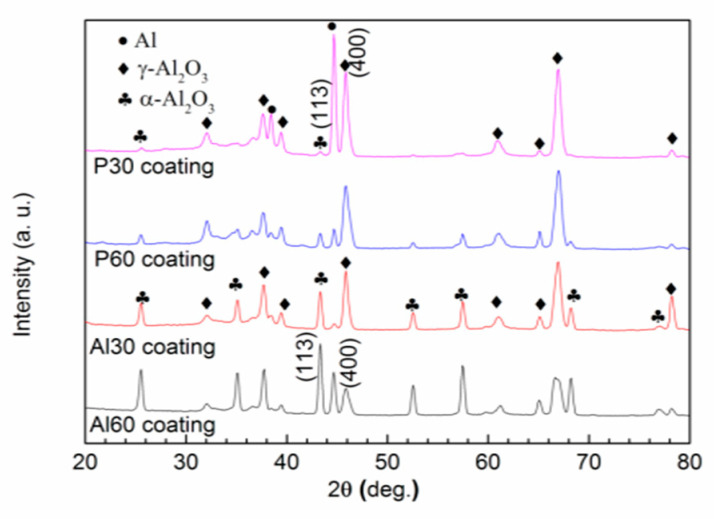
XRD patterns of the surface coatings prepared from different electrolytes with different treating time.

**Figure 5 materials-14-04037-f005:**
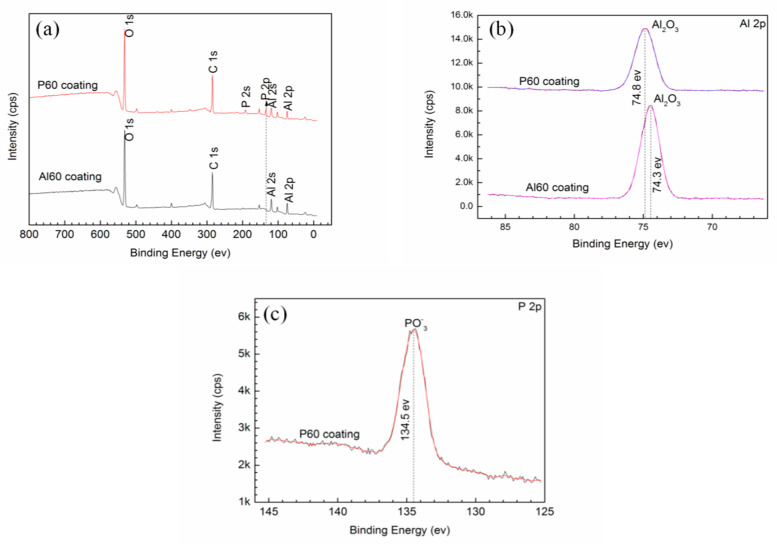
XPS spectra of the surface coatings prepared from different electrolytes with different treating time. (**a**) The surveys of the A60 and P60 coatings, the high-resulotion spectra of (**b**) Al 2p and (**c**) P 2p.

**Figure 6 materials-14-04037-f006:**
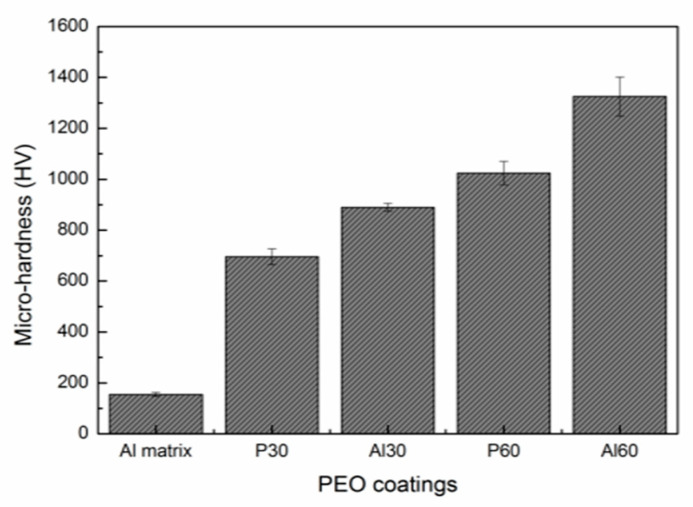
Micro-hardness of 6061 Al alloy substrate and PEO coatings.

**Figure 7 materials-14-04037-f007:**
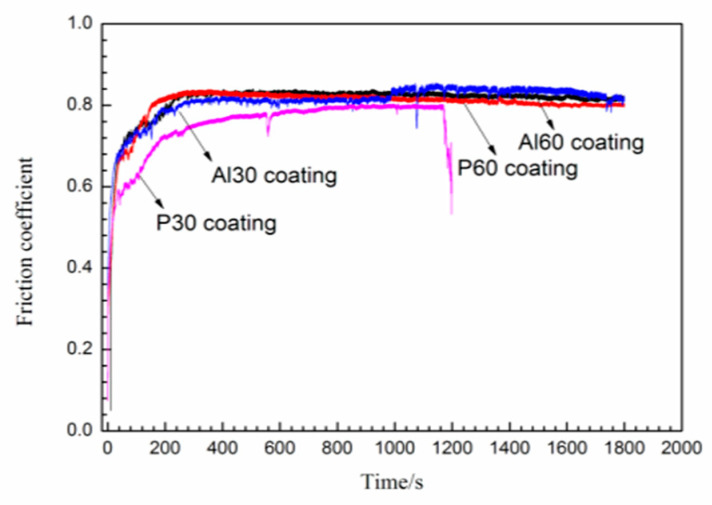
Friction coefficient as a function of sliding time under 5 N for P30, Al30, P60, and Al60 coating.

**Figure 8 materials-14-04037-f008:**
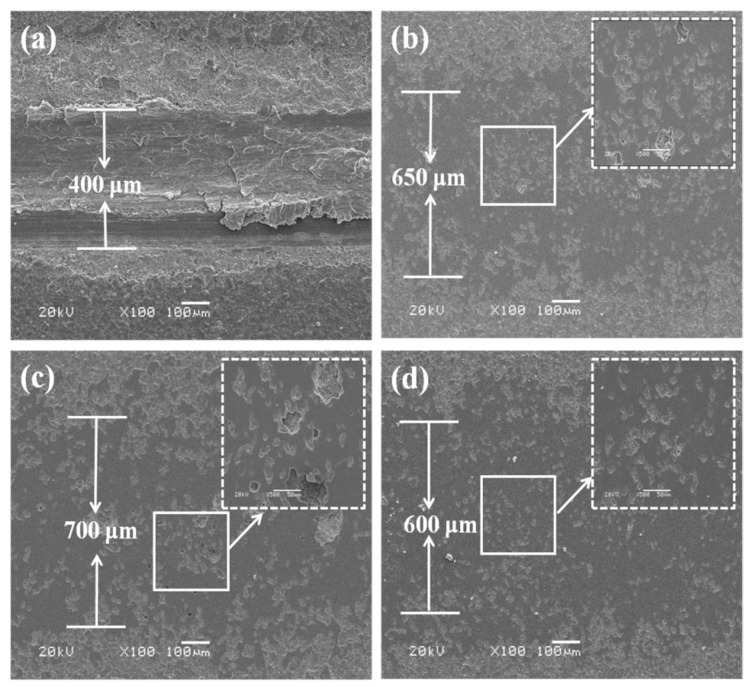
SEM images of wear marks at different coatings surface under dry friction conditions: (**a**,**b**) P30 and P60 coatings; (**c**,**d**) Al30 and Al60 coatings.

**Figure 9 materials-14-04037-f009:**
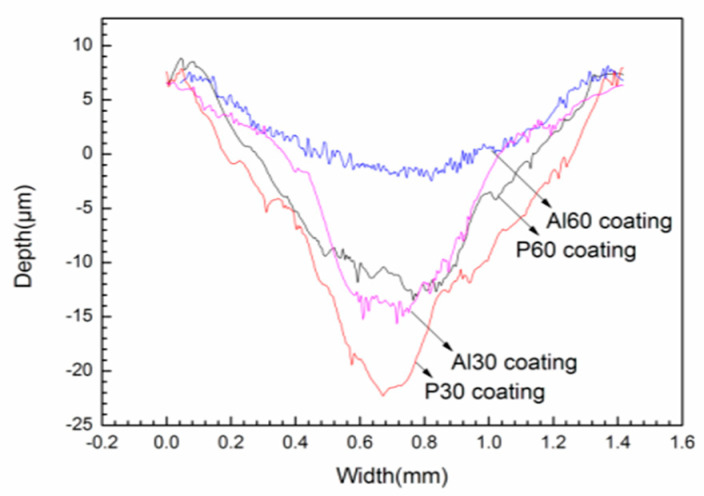
Cross-sectional profiles of wear marks at loads 5 N under dry friction conditions on the PEO coatings.

**Figure 10 materials-14-04037-f010:**
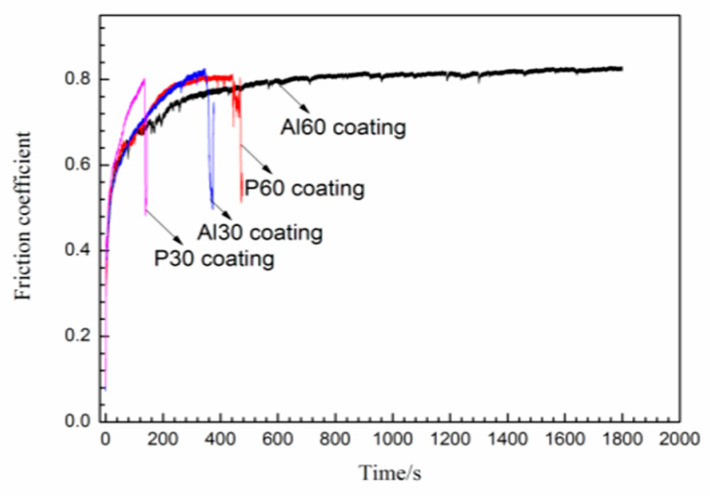
Friction coefficient as a function of sliding time under 7 N for PEO coatings.

**Figure 11 materials-14-04037-f011:**
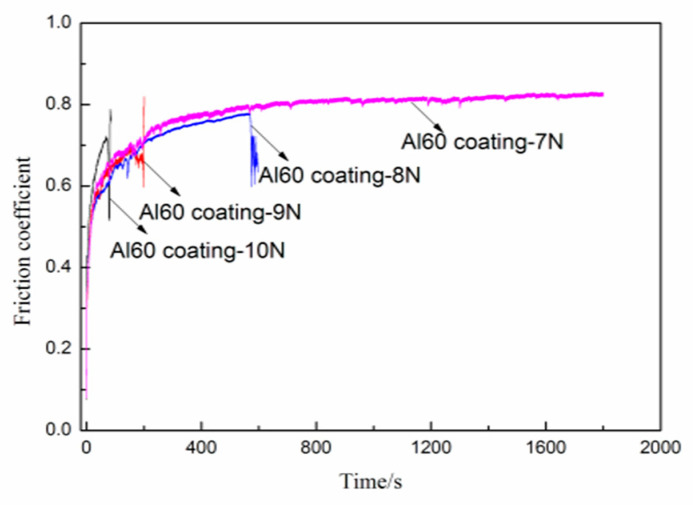
Friction coefficient as a function of sliding time under different loads for Al60 coating.

**Figure 12 materials-14-04037-f012:**
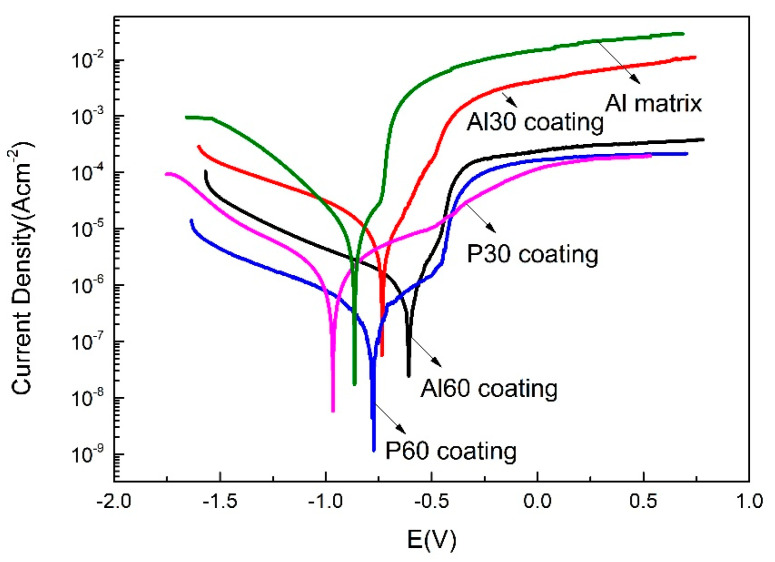
Potentiodynamic polarization curves in 5 wt % NaCl solution (pH = 3.1–3.3) for uncoated and PEO coated.

**Figure 13 materials-14-04037-f013:**
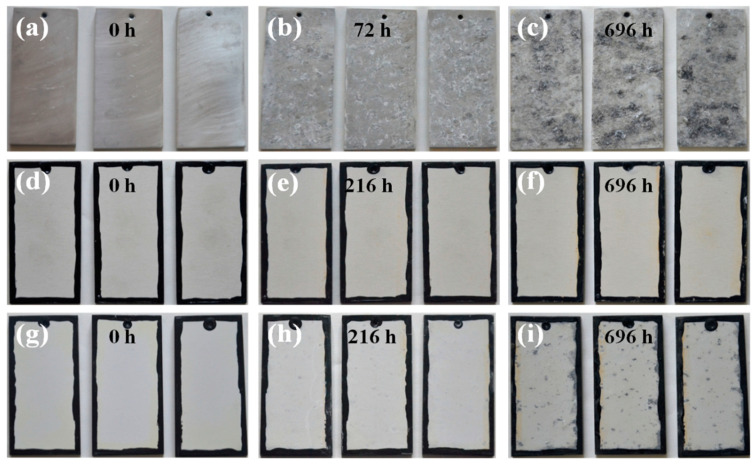
Optical appearance of the PEO coatings after the anti-corrosion property tests in acid 5 wt % NaCl solution (pH = 3.1–3.3) for 696 h: (**a**–**c**) 6061 Al alloy; (**d**–**f**) P60; (**g**–**i**) Al60.

**Table 1 materials-14-04037-t001:** The average friction coefficients and wear rates of the PEO coatings.

Coatings	Coefficient of Friction	Width of Wear Track (μm)	Depth of Wear Track (μm)	Wear Rate (mm^3^/N·m)
P30	0.78 ± 0.01	400	29	6.46 × 10^−4^
P60	0.81 ± 0.01	650	18	1.66 × 10^−4^
Al30	0.83 ± 0.02	700	20	1.37 × 10^−4^
Al60	0.82 ± 0.01	600	8	8.24 × 10^−5^

**Table 2 materials-14-04037-t002:** Fitting results of potentiodynamic polarization curves in 5 wt % NaCl solution (pH = 3.1–3.3) for uncoated and PEO coated.

Sample	E_corr_ (V vs. AgCl)	i_corr_ (A·cm^−2^)
Al alloy	−0.874	9.28 × 10^−6^
P30 coating	−0.964	1.22 × 10^−6^
Al30 coating	−0.739	4.77 × 10^−6^
P60 coating	−0.784	1.09 × 10^−7^
Al60 coating	−0.610	5.61 × 10^−7^

## Data Availability

The data presented in this study are available on request from the corresponding author.

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
