# Peer review of "Wear and Corrosion Resistance of Plasma Electrolytic Oxidation Coatings on 6061 Al Alloy in Electrolytes with Aluminate and Phosphate"

_materials, 2021, doi:10.3390/ma14144037_

Round 1
Reviewer 1 Report
Line 14 "pin on disk", but in line 87 "ball on disk" . What is correct?
Figure 2. Magnification and micron line or micron scale.
Figure 2(e, f, g h ,i) SEM picture
Line146 "3microns" Line 149 "5-10 microns" but in the figure 2(a, b, c, d) it is not possible see it
3.3 Phase and chemical composition of PEO coatings: How do you integrate the area?
Figure 4b up left (Al 2p) 4c (P 2p) what does mean?
Line 219 "16min" write it in seconds.
Have you done only one test per sample in pin on disk?
Table 1 The values in the table do not match the graph.
Figure 10 Include the 7N graph for better comparison.
Line 288 Table 2 in capital letter
Reviewer 2 Report
Authors compare PEO process on 6061 Al alloy carried out with two electrolyte compositions in respect to both corrosion and wear resistance. Results revealed that phosphate produced higher anticorrosion performance, whereas aluminate facilitated formation of wear stable coatings. This may be of interest for readers of Materials. Please find my comments below:
- Title is too broad. It should be more informative. It states the “effect of electrolytic components”, but work shows only phosphate and aluminate. I suggest following title: Wear and corrosion resistance of plasma electrolytic oxidation coatings on 6061 Al alloy in electrolytes with aluminate and phosphate. – in this case the work is absolutely clear.
- 1. line 66. What is “electromagnetic stirrer”?
- 1. Lines 63. Electrical regime requires more detailed description. Please provide examples of current-voltage waveform (real or sketch). Description of power supply should be followed by electrical regimes, rather than electrolyte compositions.
- 1. Line 67. “The two different solutions were used in the PEO process: (1) 20 g/L Na3PO4•12H2O, 1 g/L KOH with pH about 11; (2) 15 g/L Na2AlO2, 1g/L KOH with pH about10.5. “ – aluminate formula is wrong (NaAlO2). Authors probably did not calibrate pH meter since both values do not correspond to the composition of the solutions. Electrolyte (1) should have pH more than 12, electrolyte (2) more than 13.
- 2. Line 92. “5 wt% NaCl (pH=3.1-3.3)” – pH of 5%NaCl is nearly neutral. Please check for any instances in the text.
- 3. should be expanded horizontally.
Sec3.3. Line 164. “there are broad peaks between 24~ 42°” – may be hump?
